# Nano X-ray Tomography Application for Quantitative Surface Layer Geometry Analysis after Laser Beam Modification

**DOI:** 10.3390/ma15175935

**Published:** 2022-08-27

**Authors:** Aneta Gądek-Moszczak, Norbert Radek, Izabela Pliszka, Joanna Augustyn-Nadzieja, Łukasz J. Orman

**Affiliations:** 1Faculty of Mechanical Engineering, Cracow University of Technology, Al. Jana Pawła II 37, 31-864 Cracow, Poland; 2Faculty of Mechatronics and Mechanical Engineering, Kielce University of Technology, Al. 1000-lecia P.P. 7, 25-314 Kielce, Poland; 3Faculty of Management and Computer Modelling, Kielce University of Technology, Al. 1000-lecia P.P. 7, 25-314 Kielce, Poland; 4Faculty of Metals Engineering and Industrial Computer Science, AGH University of Science & Technology, 30 Mickiewicz Avenue, 30-059 Cracow, Poland; 5Faculty of Environmental, Geomatic and Energy Engineering, Kielce University of Technology, Al. 1000-lecia P.P. 7, 25-314 Kielce, Poland

**Keywords:** surface geometry, image processing, 3D analysis, laser treatment, thickness

## Abstract

Analysis of the geometrical structure of the surface of materials is an issue already widely recognised and included in international standards. The authors present the possibilities of extending the analysis of the coatings’ geometrical structure through X-ray nanotomography imaging, three-dimensional image processing, and stereological methods. Analysis of the state of the art reveals that there are no scientific rapports (indexed by Scopus) on the application X-ray nanotomography for analysis of the geometry of a coating. The presented study shows that this imaging technique can be applied and provide additional information on the quality of the layer. The comparative tests were carried out on samples with a cermet coating before and after laser treatment, including standard tests of the surface geometry and the analysis of three-dimensional images obtained from nanotomography. A set of parameters describing the compactness and thickness distribution of the applied coating is proposed, which facilitates the qualitative assessment of the application process and improvements through the additional processing of technological layers. The obtained results show that although the average thickness values before and after laser treatment did not differ significantly, their distribution on the sample surface was different, as presented in the visualisation. The determined stereological parameter *N_V_* (number of objects per unit volume) allowed for the assessment of the layer compactness, and as the analyses showed, this value decreased by more than two times after laser treatment. The analysis of the degree of sample coverage by the cermet layer showed that it increased from 70% to 95% after laser treatment, which confirms the treatment’s positive effect on the layer’s quality. This research shows that three-dimensional analysis significantly enriches the information about the geometry of the surface layer, providing data which other research techniques are unable to acquire.

## 1. Introduction

The properties of the material surface are essential for the exploitation properties of the final elements and determining their durability. Thus, correctly modelling the surface geometry is a crucial issue in surface engineering. The geometry of the material’s surface influences the exploitation properties, which is well known and widely described. The impact of the surface geometry on the tribology properties was described by Atkins [1], Maldonado-Cortés et al. [2,3], and Holmberg et al. [4,5]. In their manuscript, Chyr et al. presented the results of a study on the influence of the geometry of surface on the friction and increasing the longevity of prosthetic hip joints [6]. Davis [7] described how applying the surface engineering methods improves the corrosion and wear resistance of the materials.

Applying a surface layer on the base material is a method for enhancing the properties of the elements and obtaining specifications for the application surface of manufactured components. Enhancement of the metal surface’s exploitative properties on metallic alloys may be obtained by different methods: mechanical, thermo-mechanical, thermal, thermo-chemical, electrochemical and chemical, and physical. Each method of producing the surface layer allows for obtaining a specific type of surface layer with a particular thickness and purpose. The same methods can be carried out using different processes. All those methods can produce both protective coatings and top layers, except for mechanical methods used only for manufacturing top layers. Kovalchenko et al. [8] proved the effectiveness of laser texturing of steel surfaces and the speed-load parameters for lubrication regime transitions from boundary to hydrodynamic. Manish [9] presented a complex overview of surface engineering methods for enhancement against wear. The studies of Goud [10], Reynolds et al. [11], Radek et al. [12], and Burkov [13] proved the positive impact of surface modification by electro-spark deposition.

In production, the practice surface layer may be deposed by applying a broad spectrum of technologies, including thermo-chemical surface layer modification processes such as nitriding, physical vapour deposition (PVD), and chemical vapour deposition (CVD), as well as electroplating to improve wear or corrosion resistance. The electro-spark deposition (ESD) method belongs to the group of techniques based on using a concentrated energy stream. It is a process that can reduce component wear, improve the coating’s resistance to corrosion, and extend the exploitation time of the component or tool. B. R. Lazarenko and N. I Lazarenko developed the process that Gould [12] and Reynolds et al. [13] described in their papers. Electric sparks deposit the wear electrode material on the element in this method. When the capacitor’s energy is released, a direct current produces a high-temperature plasma arc (8000–25,000 °C) between the electrode tip and the workpiece. The plasma arc ionises the consumable material, and a small amount of the molten electrode material is transferred to the workpiece. The electro spark deposition method is cheap and high in energy, adjusting the surface to resist wear and corrosion. The advantages of this method led to the broad application of ESD coating for ship propeller components, casting moulds, fuel supply system components, and exhaust system components to protect new elements or recover the properties of worn parts [12,13,14].

The formation of the anti-wear carbide-ceramic coatings with electrodes made by powder metallurgy methods is effective and low in cost. Super hard coatings can be applied to the cutting edges of tools such as turning knives, cutters, chisels, and taps. Considering the properties of coatings deposed by the ESD method, it can be successfully used on machine elements that operate in extreme conditions, such as intense abrasive wear and impact loads, as proven by the research of Radek et al. [14,15].

The ecological thread is another advantage, encouraging the use of super hard spark erosion coatings. Holmerg and Erdamir [5] published an analysis on the influence of the ESD method on global energy consumption, cost, and emissions, and based on their work, it can be concluded that no harmful effects on the environment were identified for EDM treatment.

In the ESD method, some failures in the coating appear that prevent its broader application in production in more well-recognised areas. Significant imperfections include a high surface roughness and cracks in the layer, because the roughness becomes so high due to the treatment that it limits its use. At the same time, the residual stresses cause cracks in and brittleness of the layer and reduce its quality. The most common defects of this type of coating were identified in several studies conducted by Gould [12], Reynolds et al. [13], and the research group of Radek [14,16,17,18].

One of the essential disadvantages of ESD coatings is their high final roughness, and this is crucial when applying coatings to very smooth surfaces. Depending on the discharge energy, surface roughness heights are 2–20 µm. The analyses performed [16,17] confirmed that the shaping of the surface occurs due to the overlapping of craters resulting from the erosion of the substrate, as well as ridges caused by particles of the electrode coating material moving to the surface.

According to Radek et al. [16,18] and Pietraszek et al. [19], due to reducing the roughness of ESD coatings, laser modification may be applied. The growing interest in the radiation emitted by the laser results from its specific beneficial properties and the possibility of building radiation sources with various parameters, such as the wavelength, beam transverse mode, radiated power, pulse energy, pulse duration, and pulse repetition frequency. The properties of the processed material significantly influence the result of the laser modification. The most important of them are the density, heat capacity, enthalpy, specific thermal conductivity, thermal diffusivity, and the material absorption coefficient for laser radiation.

The material absorbency depends not only on the wavelength of the laser radiation but also on the method of surface preparation and the temperature of the processed elements, as well as the time of the laser radiation’s impact on the material.

Laser treatment eliminates the surface of ESD coating defects such as scratches, delamination, and microcracks. The primary purpose of laser processing of layers is to improve their quality and homogenise their structure and chemical composition [15,16,18,20,21].

Considering the properties of the coating deposed by the ESD method and modelling the geometry by laser treatment, this type of specimen is a suitable example for analysis with the employment of three-dimensional nanotomography and three-dimensional image processing, as well as analysis techniques that were identified in further research by Gądek-Moszczak et al. [22,23].

The main research goal was to analyse the possibility of using three-dimensional image data for the quantitative and qualitative description of the surface layer geometry. The indirect goal was to develop an appropriate coating detection method so that the image intended for measurement would reflect the actual structure as accurately as possible. It was equally important to determine a set of parameters that would be sensitive to changes in the spatial geometry of the layer after laser processing, among the many available parameters for the geometric description of objects.

A thorough analysis of the literature revealed a lack of scientific papers (indexed in global databases) on the use of X-ray nanotomography to analyse the geometry of the coatings.

Based on the obtained visual data in the form of three-dimensional reconstructions and the results of quantitative analysis, an evaluation of the usefulness of the proposed three-dimensional analysis method for qualitative and quantitative assessment of the surface layer geometry was carried out.

## 2. Materials and Methods

### 2.1. Material and Treatment Parameters

The base material used in the tests was carbon steel C45, which showed sufficient corrosion resistance in oxidising atmospheres up to approximately 500 °C. The permissible operating temperature of the C45 steel structure parts depended on the corrosion resistance under the given conditions, the time of exposure to high temperatures, and the size and frequency of its fluctuations. C45 steel is carbon steel from which machine parts are made (e.g., body moulds for plastic processing, crankshafts, gears, bushings, thrust washers, rollers, pump impellers, screw discs, or wheel hubs). This durable steel with high ductility is widely used in industries but is characterised by poor machinability, resulting in high roughness of the surfaces. The widespread use of C45 steel can be found in the machine industry, where it is used for producing, for example, gears, spindles of lathes and drills, connecting rods, piston rods, turbine shafts, shafts, and blades [18].

To enhance the properties of the surface layer of the C45 steel specimens, the cermet coating was chosen. The use of ceramic tool materials compared with sintered carbides is small but still shows dynamic growth. According to estimates, about 5% of cutting tool blades are made of this group of materials. The most popular materials for the production of ceramic tool materials include the following:Single-phase Al_2_O_3_ aluminium oxide;Silicon nitride Si_3_N_4_;Multi-phase mixtures of Al_2_O_3_ and Si_3_N_4_ with hard carbides, nitrides, and oxides.

The research was conducted on the specimens with the WC-CO-Al_2_O_3_ coating obtained by the electro-spark deposition (ESD) method. The Kielce University of Technology led the process of preparing the samples. For comparative analysis of the geometry of the surface changes, two types of probes of hardened C45 steel with the WC-Co-Al_2_O_3_ coating were used. The first set of probes had an WC-Co-Al_2_O_3_ layer, no additional treatment, and laser treatment afterward.

The process of deposition by the ESD method was performed using EIL-8A equipment (Ukrainian production). Due to producer indication as well as the results of previous experiments, the parameters of the ESD process were as follows:Voltage (U): 230 V;Capacitor volume (C): 300 μF;Current intensity: (I): 2.2 A;Exposition time (T): 2 min/cm^2^.

After deposition of the cermet coating to enhance the surface geometry, the second type of analysed sample was treated with Nd via a YAG laser (impulse work), model BLS 720. Laser treatment should result in surface roughness reduction and changes in the shape of the profile irregularities. The purpose of laser compaction is to reduce the porosity of the coating and eliminate defects such as scratches, delamination, and cracks in the coating. After laser treatment, the surface layer is expected to have improved integrity, which strongly impacts the layer’s mechanical and functional properties. Proper adjustment of the laser beam is also an important issue because, for smoothing, a low power density and large-diameter laser beam melts only the tops of the asperities. Laser treatments were conducted with the following parameters:Spot diameter (d): 0.7 mm;Power (P): 20 W;Laser beam velocity (v): 250 mm/min;Nozzle-workpiece distance (∆l): 1 mm;Pulse duration (t): 0.4 ms;Pulse repetition frequency (f): 50 Hz;Beam shift jump (S): 0.4 mm;Nitrogen gas yield (Q): 25 L/min.

### 2.2. Scanning Microscopy

Microstructure analysis is a standard test of quantitative and qualitative analysis of materials, including materials with deposited coatings. A high magnification of microscopy observation allows noting possible defects in the coating. The structure of the tested materials was assessed on specimens made in a plane perpendicular to the applied surface layer, which made it possible to determine the coherence of the layer with the substrate and measure its thickness. The observation was performed on a scanning microscopy Joel type JSM-5400 with a magnification of 1000×.

### 2.3. X-ray Diffraction

An X-ray diffraction study was performed at AGH University of Science & Technology, a PHILIPS X-Ray Diffractometer PW 1830 utilising the X-ray diffraction technique was applied to determine the coatings’ phase compositions. The Cu anode lamp provided filtered Kα radiation. The voltage supplied to the device amounted to 40 kV, while the current was 30 mA. The measurements were carried out at 2θ from 30° to 60°. The speed of scanning was set to 0.05°/3 s.

### 2.4. Profilometry

The tests aimed at obtaining a geometric surface structure and roughness profiles were conducted at Kielce University of Technology (in the Geometric Measurements Lab.). An optical profilometer called Talysurf CCI was applied. The testing field had an area of 1.65 × 1.65 mm^2^, while the horizontal resolution amounted to 1.65 × 1.65 µm^2^. TalyMap Platinium (ver. 6.2) software was applied to analyse the surfaces’ 3D images. Thus, the geometric surface structures of the layers might have been determined.

### 2.5. X-ray Nanotomography

The three-dimensional imaging technique using X-ray microtomography is an increasingly popular method of examining the micro- and nanostructures of materials in different fields of science, such as material engineering, biomaterials, and biology [24,25,26,27,28,29]. Its undoubted advantages include the accurate representation of the spatial structure of the material without destroying it and with no need for special preparation, as is the case with other imaging techniques, such as microscopy observation. The sample could be reused for imaging under different exposure conditions or further experiments. X-ray nanotomography is used to analyse materials with complex spatial structures with a spatial resolution of 1 μm or less. A description based on two-dimensional cross-sections does not fully reflect the existing spatial relations between its components. Quantitative analysis of the three-dimensional structure has been widely used in materials such as metallic or ceramic foams and composite materials, where the description of the spatial arrangement of reinforcing particles or fibres significantly impacts the material’s strength properties. Three-dimensional analysis of the surface layers is not widely used. Still, it could provide information complementary to standard two-dimensional structural tests (light microscopy or SEM), which would be valuable from the point of view of production technology and deposition of the coating on the core material.

Three-dimensional analysis based on images obtained from X-ray nanotomography of the surface layer is not a study that may be performed for all materials.

This is limited to a specific combination of the base material and the type of material chosen for the coating. X-ray nanotomography is an imaging technique based on the material’s interaction with an X-ray beam. After passing through the sample, the X-rays are attenuated, and the level of attenuation depends on the thickness of the test object, the atomic number of the sample element, and the beam energy. The absorption *I* of the X-ray beam of an intensity *I*_0_, which passes through an absorbent with a thickness of *x*, can be calculated according to Beer’s absorption formula [24]:(1)$$I=I_{0}e^{-\mu x}\;$$
where *μ* is a linear absorption coefficient.

In practice, the most frequently used variable is the mass absorption coefficient *µ_m_*, which is equal to the ratio of *µ* to the absorbent density ρ. Materials containing atoms with a high *Z* atomic number absorb X-ray radiation more efficiently than substances consisting of elements with a low *Z* value. The mass absorption coefficient of *µ_m_* increases with the atomic number as approximately *Z*^3^. Taking into account this fact and the impact of the wavelength on absorption, the mass absorption coefficient is defined by the formula
*µ_m_* = *c*
*λ*^3^
*Z*^3^
(2)
where *c* is approximately constant between the different absorption thresholds.

The desirability of using X-ray micro or nanotomography to test samples consisting of several components, as is the case with porous materials, composites, or materials with coating systems, can be assessed before the test is performed by estimating the radiation absorption of each of the components. The greater the difference in the value of *I*, the better the contrast in the images, and the better representation of the individual elements of the structure of the tested sample.

Images that resulted from X-ray nanotomography were recorded as a set of bitmap images or as a RAW file. They could be used to reconstruct the three-dimensional structure in the material study.

## 3. Results and Discussion

Samples of C45 with WC-Co-Al_2_O_3_ before and after laser melting were subjected to research, providing knowledge about changes in the coating after laser processing microstructure analysis of SEM images with EDS analysis, analysis of the geometry of the surface layer using a profilometer, and analysis of the geometry of the layer on images from computed nanotomography.

### 3.1. Microstructure Analysis

Based on the SEM microstructure images, a qualitative and quantitative analysis was performed, considering the measurement of the thickness of the obtained layers or the range of the heat-affected zone (HAZ) after laser treatment. The SEM images presented the cross-section of the specimen (Figure 1), showing a clear border between the coating and the substrate. Unfavourable phenomena were observed, such as pores and microcracks in the layer. Discontinuity of the coating is an undesirable feature that significantly reduces properties such as adhesion or anticorrosion.

The modification of the WC-Co-Al_2_O_3_ coating with a laser beam homogenised its chemical composition. The technological surface layers (TWP) produced by laser melting did not have microcracks and pores (compare Figure 1 and Figure 2). The thickness of the WC-CoAl_2_O_3_ coating after laser modification changed and was in the range of 40–50 microns. The content of the HAZ was about 30–40 microns deep into the base material. The layer thickness measured for the cross-section SEM microphotography ranged from 30 to 40 micrometres.

EDS study, was conducted on the specimens after laser treatment (Figure 3), Figure 4 shows the EDS spectrum of the WC-Co-Al_2_O_3_ coating. The presence of W, Co, Al, C, and Fe proved the alloy formation between the coating and the substrate.

The X-ray diffraction tests enabled us to determine the WC-Co-Al_2_O_3_ coating’s phase composition, as presented in Figure 5. It turned out that it was mainly composed of WC as well as W_2_C phases. There were also marginal impurities of Al_2_O_3_ and Co_2_C found in the sample. The laser beam led to melting of the WC-Co-Al_2_O_3_ coating into the base substrate. The occurrence of new Fe_3_W_3_C and Fe phases (next to the WC and W_2_C phases) was also observed. The highest peaks in Figure 5a,b can be attributed to W_2_C and Fe_3_W_3_C, respectively.

### 3.2. Surface Geometric Structure and Roughness Measurements

It is considered that surface geometric structure (SGS) parameters significantly determine the characteristics of a given surface. They tend to have a significant impact on several phenomena which occur within the considered layer.

SGS tests are applied in practice to generate images that properly understand surface characteristics. The following steps are considered if the surface’s geometry features need to be determined: measurements conducted with a selected technique, surface presentation, and its parametric assessment. Figure 6 presents the surface topography of a WC-Co-Al_2_O_3_ coating before and after laser beam machining (LBM).

Table 1 presents the most important parameters of the surface geometric structure of the tested samples and their averaged values from 10 measurements.

The C45 carbon steel samples on which the coatings were applied were characterised by the value of the parameter *Sa* = 0.74 ÷ 0.82 µm. The *Sa* parameter is the essential amplitude parameter for the quantitative assessment of the condition of the analysed surface. The laser treatment lowered the *Sa* parameter of the WC-Co-Al_2_O_3_ coating, which proved that we were dealing with laser smoothing (reduction of the height of the tops). The WC-Co-Al_2_O_3_ coatings were characterised by the value of the parameter *Sa* = 3.43 − 3.87 μm, while after laser beam machining, the arithmetic mean of the surface height ranged from 3.79 to 4.11 μm.

The arithmetic mean of the surface height of the coating (*Sa*) (i.e., the essential amplitude parameter for the quantitative assessment of the condition of the analysed surface) decreased by about 8% after laser treatment compared with the coating without this treatment. The measurements show that after the laser treatment, there was a slight decrease in the *Sa* parameter of the WC-Co-Al_2_O_3_ layer. The smaller value of the *Sa* parameter resulted from the tensile forces acting on the surface and, accordingly, the motion of the liquid metal. As a result, a homogeneous surface profile was obtained after solidification. It was also found that the laser smoothing method allowed for achieving the directivity of the micro-geometry structure at the same time. If pulse laser treatment is applied, it is assumed that the main factor affecting the surface profile after solidification is the pressure of vapour causing the disposal of the material from the central zone and the production of characteristic flashes on the boundary between the melted and unmelted zones.

A similar tendency was observed for the mean square surface height *Sq* and the following parameters: maximum peak height (*Sp*), maximum pit height (*Sv*), and maximum height (*Sz*).

Supplementary information on the shape of the surface of the tested samples before and after laser treatment is provided by the amplitude parameters: the skewness coefficient-asymmetry *Sku* and the concentration coefficient of kurtosis *Ssk*. These parameters are sensitive to the occurrence of local elevations or depressions on the surface, as well as defects (e.g., scratches and delamination). Positive values of the surface asymmetry *Ssk* (skew) for the WC-Co-Al_2_O_3_ coating before and after LBM provided information that we were dealing with a smooth surface without plateau-shaped elevations.

The obtained values of the surface inclination close to each other and within the range *Sku* = 4.24 ÷ 7.20 prove that the ordinate distribution for all samples was close to the normal distribution.

The WC-Co-Al_2_O_3_ coating before and after laser beam machining had a random isotropic structure within the range of *Iz* = 62.99 ÷ 69.62%.

The roughness measurement is a standard surface geometry analysis specified in standards PN-EN ISO 4287:1999 and PN-EN ISO 11562:1998. The tests were carried out for the coatings mentioned above in two perpendicular directions. The first measurement was made following the movement of the electrode, while the second measurement was vertical to the scanning stitches. The mean value of the *Ra* parameter for a given coating was calculated from two sizes.

Measurements of the laser-modified WC-Co-Al_2_O_3_ coatings were made in the direction perpendicular and parallel to the axis of the paths created with the laser beam. The average roughness value for a given layer was calculated. The *Ra* for the WC-Co-Al_2_O_3_ coating was from 3.12 to 3.35 microns, and after laser irradiation, it was from 2.71 to 2.96 µm. The coated C45 steel samples were roughly *Ra* = 0.41 − 0.43 µm. Examples of measurement charts of the microgeometry parameters of the tested samples are shown in Figure 7 and Figure 8.

### 3.3. Quantitative Analyses of Nanotomography Images of Specimens

The procedure for quantitative analysis of digital images consists of several stages, from image acquisition to 3D visualisation for the assessment of the image quality and visibility of the tested structures and image preprocessing, which includes geometrical translation and filtering, detection, measurements which provide quantitative results essential for qualitative assessment, and conclusions (Figure 9).

#### 3.3.1. Image Acquisition

The imaging study verifying the effect of laser treatment on the electro-sparking coating’s geometry was performed with Nanotom 180N nanotomography by GE Sensing & Inspection Technologies Phoenix|X-ray Gmbh. The operating parameters of the X-ray tube during the acquisition were as follows: *I* = 170 μA and *V* = 130 kV. The total acquisition time for one sample was approximately 400 min. Reconstruction of the measurement results in the form of a 3D image was carried out using GE software datosX ver. 2.1.0 with the Feldkamp algorithm for a conical X-ray beam. The final spatial resolution was 1 μm, and the dimensions of the tested samples were 2.2 mm × 3.1 mm × 3.0 mm. The resulting sequence of three-dimensional images of the tested samples was saved as an 8-bit RAW file (8 bits for 1 voxel), which gave 256 shades of grey to represent the specimens. The accuracy of the spatial projection of the generated images was 1 μm.

The visibility of the components was sufficient on the obtained images of the cross-section (stack of images from nanotomography) (compare Figure 10a,b) and allowed performing automatic and entirely objective quantitative analysis of the coating.

Images written by the acquisition system in RAW format were imported to Fiji software, an open-source platform for three-dimensional image reconstruction, visualisation, processing, and analysis [30].

#### 3.3.2. Preprocessing of Three-Dimensional Images

The first stage of assessing the quality of the layer was performed on the three-dimensional visualisation. As shown in Figure 11, when presenting the specimen with an ESD coating before laser treatment, a deposed layer did not cover the steel surface thoroughly, and the thickness of the coating was significantly uneven.

Visual assessment of a three-dimensional reconstruction of the specimen with ESD cermet coating after laser treatment confirmed the benefits of laser treatment (Figure 12). The cermet coating covered almost the whole analysed surface.

Applying automatic algorithms for quantitative analysis of the coating required detection of the layer. Detection is a process of transforming images in greyscale into binary images [31,32]. It is one of the most critical transformations in the processed procedure, because the correct selection of the method and its parameters determines the degree of compliance of the detected areas with the objects being analysed. Detection quality is a critical factor influencing the reliability of the digital measurements of the objects carried out at the final stage of the image analysis procedure. Binarisation transforms the greyscale image, which is represented by 256 intensity values, into a black and white image which consists only of two values of intensity, 0 and 1, where 0 is black and 1 is white. Calibration of the binarisation parameters is aimed at obtaining an image in which the value of one will be taken by pixels representing objects in the digital image and zero corresponds to the remaining areas of the image, which were not crucial from the point of view of the analysis. The most common detection method is thresholding, based on setting the limit values of the grey intensity, called binarisation thresholds, that separate the set of pixels representing objects from the background. This transformation is carried out by determining the value of the binarisation thresholds. Global threshold methods are based on the analysis of the image’s histogram. The optimal threshold is determined by optimising some criterion function obtained from the image’s histogram.

Let *f*(*x,y,z*) be a grey value of the voxel located at the point (*x,y,z*). Suppose the voxel intensity value on the initial image *f*(*x,y,z*) is lower than the defined threshold value *T*. In that case, it will be converted on the binary image *g*(*x,y,z*) into the value zero, and voxels whose intensity values are higher than the set threshold *T* will be converted into one, as defined in Equation (3):(3)$$g\left(x,y,z\right)=\{\begin{array}{c} 1,\;if\;f\left(x,y,z\right)>T\\ 0\;otherwise \end{array}$$

This method is very effective for high-contrast images, where the difference between the values corresponding to the object and background are significant.

The quality of the obtained images of the base material and cermet coating from X-ray nanotomography was acceptable, and the contrast between the based material and coating was well distinguished. Automatic algorithms performed detection of the analysed coating to ensure the repeatability and objectivity of the detection and further analysis. Due to reducing the number of eventual artefacts in the binary images and improving the detection result, three-dimensional median filtration was performed. Median filtration is one method of ranking filtering which effectively reduces the noise ratio in the images. In this filtration, each voxel is analysed with its neighbourhood. All considered voxel intensity values are ordered into a descending sequence of values. The value in the sequence’s middle becomes the new value of the analysed voxel in the resulting image [31,32]. The advantage of this noise reduction method is that new values are not introduced to the image, and the second important issue is that the noise is significantly reduced, but the edges of the objects remain sharp.

#### 3.3.3. Detection Methods

Cermet coating detection was possible by automatic methods chosen based on visual observation of their accuracy in surface layer detection. Among the available algorithms for automating thresholds [32,33,34,35,36,37,38], RenyiEntropy was selected because only this counted out the threshold values with acceptable accurateness of detection of the coating area, represented by red highlighted areas (Figure 13 and Figure 14).

For detection of the cermet coating for both types of specimens, an algorithm was chosen which, in this case, gave the best detection result among the other testing methods, such as Li, Otsu, maximum entropy, and Yen, which are available in the Fiji algorithm’s library [38]. The visualisation of detection effects is presented in Figure 13 and Figure 14. As can be observed, the acceptable detection results formed for both types of specimens were obtained by Otsu, Yen and RenyiEntropy. Due to a more effective adjustment to the smslightluctuation of the voxel values, for detection procedure was chosen RenyiEntropy method.

The RenyiEntropy algorithm took its name after Alfréd Rényi, who tried to work out the most general definition of information measures that preserve additivity for independent events [36]. Renyi’s entropy method is a type of generalised Shannon entropy where the coefficient α is introduced. For α = 1, Renyi’s entropy is equal to Shannon’s entropy. For a single threshold in Renyi’s entropy, the threshold *T* divides the image *I* into two parts: the object and background. The object and background grey probabilities are calculated, which are applied in Renyi’s entropy formula. According to Liu et al. [36], the method of Renyi’s entropy with a single threshold can be defined as follows. Let *I* be an image with objects which are the aim of quantitative analysis. Thus, *I* will be segmented with *L* levels. The probability of each grey level distribution is {*p*(1)*, p*(2), *p*(3), …, *p*(*L*)}. Suppose there is only a single threshold *T* to divide image *I* into two parts: the objects *A*_0_ = {1, 2, 3, …, *t*} and the background *A*_1_ = {*t* + 1, *t* + 2, *t* + 3, …, *L* − 1}. The object and background grey probabilities are calculated, where *w*_0_(*t*) is the probability of occurrence *A*_0_ and *w*_1_(*t*) is the probability of occurrence *A*_1_. The sum of both calculated probabilities is *w*_0_(*t*) + *w*_1_(*t*) = 1. The probabilities *w*_0_(*t*) and *w*_1_(*t*) are defined as follows:(4)$$w_{0}\left(t\right)=\sum_{i=1}^{t}p\left(i\right)$$
(5)$$w_{1}\left(t\right)=\sum_{i=t}^{L-1}p\left(i\right)$$

Renyi’s entropy of the image background and object is defined in the following equations:(6)$$Rt_{\propto }^{0}\left(t\right)=\frac{1}{1-\propto }\ln\sum_{i=1}^{t}{\left(\frac{p\left(i\right)}{w_{0}\left(t\right)}\right)}^{\propto }$$
(7)$$Rt_{\propto }^{1}\left(t\right)=\frac{1}{1-\propto }\ln\sum_{i=t+1}^{tL-1}{\left(\frac{p\left(i\right)}{w_{1}\left(t\right)}\right)}^{\propto }$$

The total Renyi’s entropy is defined as
(8)$$T\left(t\right)=Rt_{\propto }^{0}\left(t\right)+Rt_{\propto }^{1}\left(t\right)$$

Finally, the optimal threshold *T* should fulfil Equation (9):(9)t=argmax(T(t))

According to the Renyi’s entropy method of a single threshold, Renyi’s entropy of multiple thresholds may be worked out.

Binary images due to minimalisation of the artefacts created by the thresholds and local inhomogeneity of grey level values of coating representations were processed by three-dimensional opening (morphological transformation) [31,32] through a structural element size of one (cube-sized, 1 × 1 × 1 voxels). The space is a transformation based on two basic morphological transforms: erosion and dilation. Erosion can be described as removing the external, one-voxel-thick layer from binary objects. Dilation is the opposite of erosion. Due to the dilation layer, a one-voxel-thick layer is added to the objects in the binary images. The opening removes minor separated artefacts from the picture, and the object’s surface is smothered. This operation does not affect the geometrical properties of the analysed objects, causing the changes to have a three-voxel local range. The opening transformation corrects the boundary of the objects by smoothing them and deleting the small objects which appeared on the binary image as an effect of uneven object representation (local fluctuation of grey level values).

#### 3.3.4. Quantitative Analysis

Three-dimensional analysis of the geometrical structure involves the study of its consistency and thickness. The research was performed in Fiji software, employing the algorithms for three-dimensional analysis and applying stereological parameters to assess the crucial features of the layer [39,40,41,42].

The first parameter chosen for analysis was the total volume *V*, which gives the volume of the detected coating. The volume is assessed directly from the number of voxels representing the detected object. The second of the parameters chosen for analysis was the surface area *S*. The algorithm for measuring the surface area of three-dimensional objects creates a mesh from the detected object in the image and calculates the area of the surface of the mesh. A mesh is a set of triangular faces that defines the shape of an object in 3D graphics [42,43].

The ratio of the surface area *S* to the total measured volume *V* indirectly provides information about the degree of surface development. Analysis of the layer located in the direct location of the core material will give information on the percentage of the area of the steel surface not covered by the coating.

The number of objects *N* forming the coating, its volume, and the surface area were analysed, and the results are presented in Table 2. The number of objects in a unit of volume in both types of samples informs how consistent the coating was. The decreasing number of objects per unit of volume *N_V_* for the analysed coating confirmed that the laser treatment of the surface cermet layer deposited by ESD increased its consistency. The *N_V_* for the coating after LBM was reduced by more than 70% compared with the sample without the LBM.

The total volume *V* of the coating after LBM increased, which was an effect of the alloying properties of the laser treatment that caused melting of the cermet layer and the core material, which is a positive phenomenon because it increases the coherence of the coating with the base material.

The surface area *S* is a feature that may be used in this analysis to assess the changes in the surface geometry into a more complex one textured by laser machining. This effect was confirmed by the roughness measurement, where *Ra* increased from 3.12 to 3.35 µm, and after laser irradiation, it increased from 2.71 to 2.96 µm. The value of the ratio of the area surface and total volume provided information on the complexity of the surface. In general, two objects with the same total volume and different *S/V* values differ by the complexity of their shapes. One of the commonly used shape factors for two-dimensional objects is the ratio of the perimeter and area of the objects [30]. Parameter *A_A_* is an area fraction of the sample surface covered by a WC-Co-AL_2_O_3_ coating. This is calculated by dividing the total surface area covered by a coating by the total surface area of the sample. As presented in Table 2, it increased from 70% to 95%, which confirms the positive impact of laser machining on the quality of the ESD coating.

The results of stereological analysis of the three-dimensional images of a coating provide the quantitative description of the coating, which is unique, valuable, and essential for reliable analysis of the influence of the technological process and its adjustment on the quality of the layer.

The thickness was measured by applying the algorithm for the model-independent thickness assessment in three-dimensional images [40]. The measured local thickness was assessed as a value of the diameter of the enormous sphere inside the object and contained the point of the object’s boundary.

As a result of the analysis, the value of the minimal, maximal, mean, mode, and standard deviation thickness is provided (Table 3). The three-dimensional chart presents the thickness of the layer in all analysed areas (Figure 15, Figure 16 and Figure 17).

Figure 15 and Figure 16 provide essential information on the thickness distribution in the analysed specimen. When analysing the thickness map, it is evident that the surface layer deposited by the ESD method was highly heterogeneous. There was a region where there was a lack of coating, and there were regions where the thickness of the layer was several times higher than in the other areas.

The effect of the laser treatment can be observed on the thickness map. Although the mean value of the thickness was close to the mean value for the sample without the laser treatment (see Table 3), the thickness map and the three-dimensional chart of the thickness layer on the analysed volume show that the distribution of the thickness was more homogeneous (Figure 18 and Figure 19). The histograms of the thickness distribution also present significant differences (Figure 20). The values of the basic statistic for the samples before and after laser treatment were similar except for the mode, which after laser machining was significantly lower. The results of the statistical analysis without the graphic visualisation of the thickness distribution did not provide enough information to accurately describe the changes. Combining two sources of information—visual and quantitative—allowed better assessment of the changes in the coating introduced by laser machining.

## 4. Conclusions

The laser treatment eliminated deposition imperfections (pores, cracks, etc.) and slightly changed the chemical and phase compositions of the coating. The surface layer of the WC-Co-Al_2_O_3_ coatings consisted mainly of the W_2_C and WC phases, while after laser beam machining, it mainly consisted of the Fe_3_W_3_C and W_2_C phases.The parameters of the surface geometric structure of the ESD WC-Co-Al_2_O_3_ coatings after laser treatment had lower values by approximately 7% compared with the coatings before this treatment.Analysis of the images from X-ray nanotomography of a cermet coating provided a valuable opportunity to analyse a sample with a coating in a specific volume without the bias of sampling error or resulting from the preparation of the sample for microscopic observation. This additional quantitative information, supported by three-dimensional visualisation, allowed for a more precise assessment of the quality of the applied coating and the impact of a different technological process, such as laser processing of its geometry and not just the surface.Three-dimensional analysis provided additional parameters characterising the coating geometry, including compactness and thickness, which are complementary to other standard research methods.Further tests should include measurements of the internal stresses and tests of the tribological electro-spark coatings before and after laser treatment.

## Figures and Tables

**Figure 1 materials-15-05935-f001:**
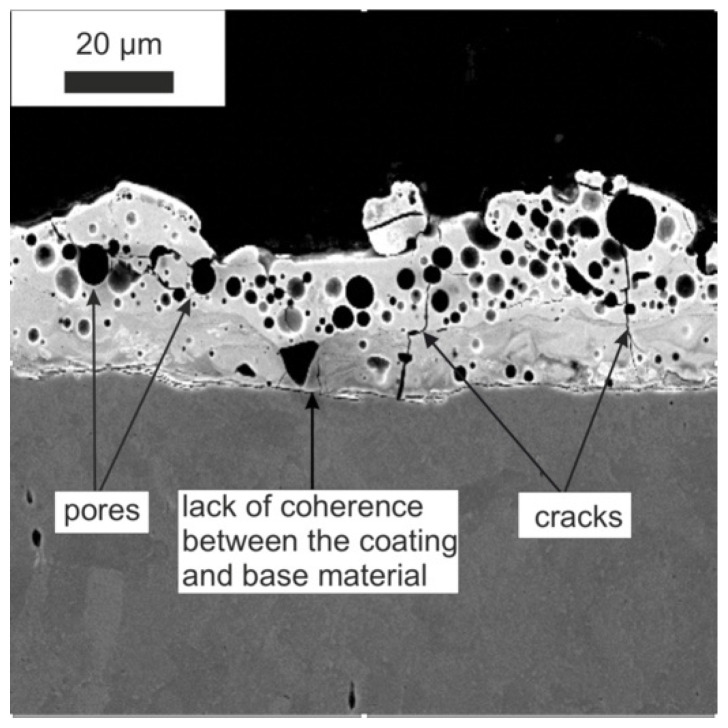
The microstructure of the WC-Co-Al_2_O_3_ coating before laser treatment (over 1000x).

**Figure 2 materials-15-05935-f002:**
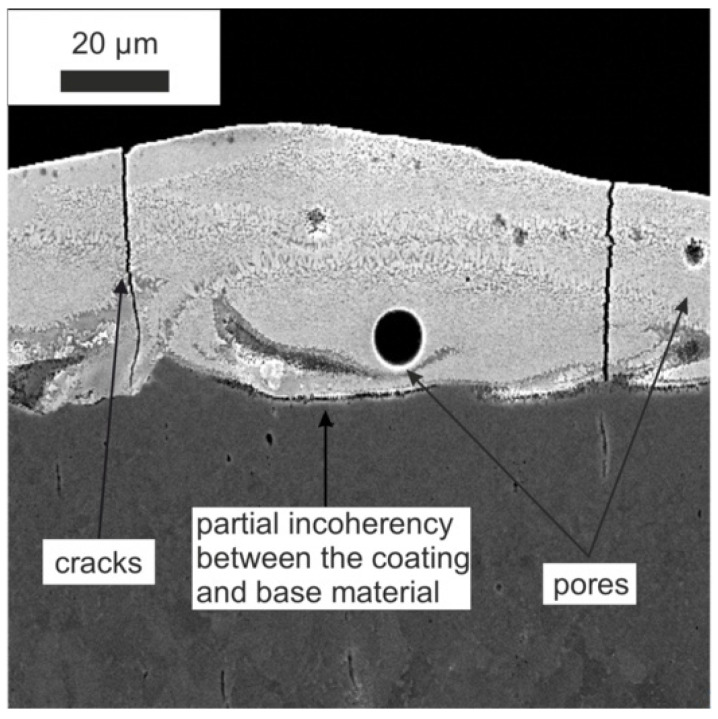
The microstructure of the WC-Co-Al_2_O_3_ coating after laser processing (over 1000x).

**Figure 3 materials-15-05935-f003:**
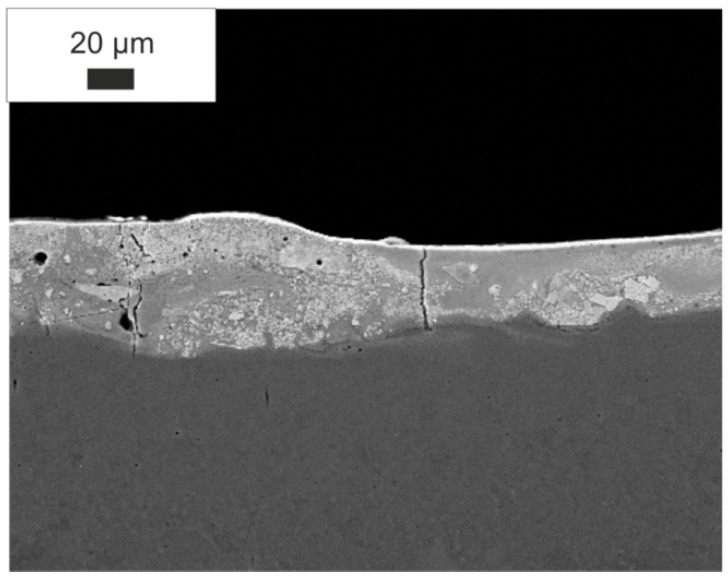
Microstructure of the WC-Co-Al_2_O_3_ layer after laser treatment.

**Figure 4 materials-15-05935-f004:**
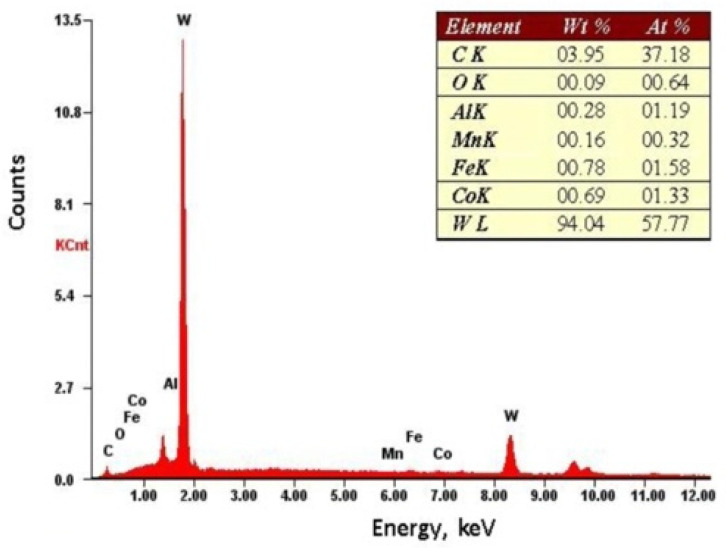
The spectrum of the characteristic X-ray radiation for the WC-Co-Al_2_O_3_ shell.

**Figure 5 materials-15-05935-f005:**
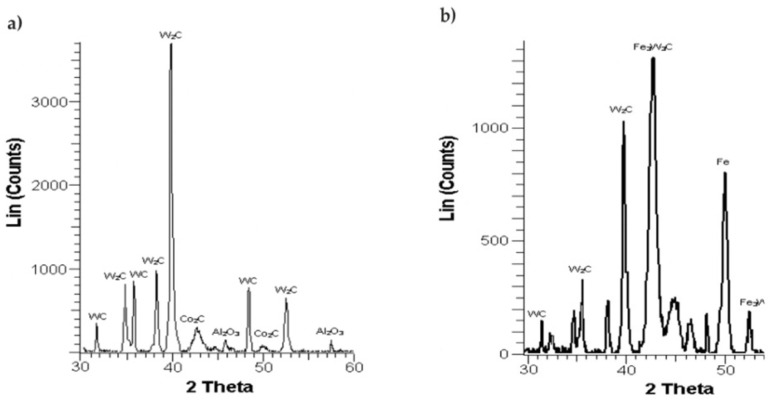
WC-Co-Al_2_O_3_ coating’s diffraction pattern (**a**) prior to laser beam processing and (**b**) after laser beam processing.

**Figure 6 materials-15-05935-f006:**
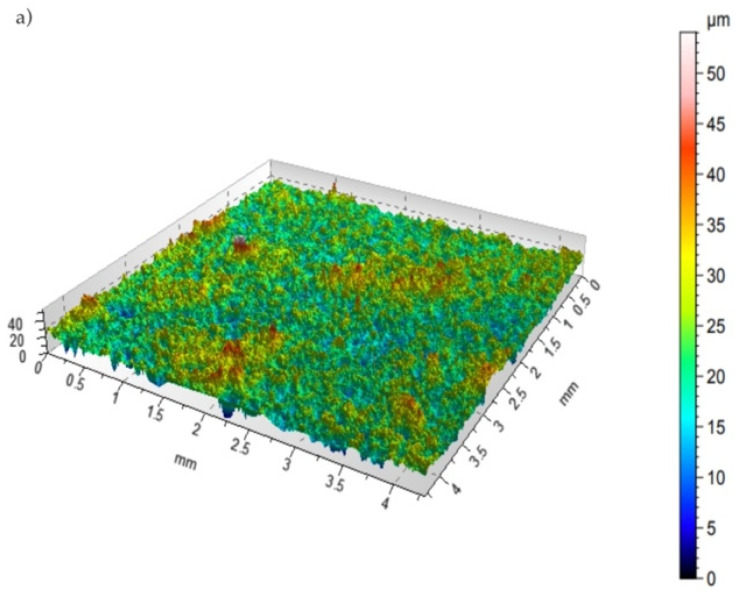
Topography of coatings (**a**) before LBM and (**b**) after LBM.

**Figure 7 materials-15-05935-f007:**
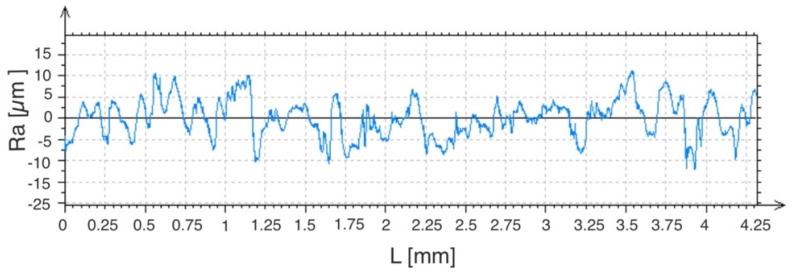
Examples of measurement results of the WC-Co-Al_2_O_3_ coating microgeometry parameters.

**Figure 8 materials-15-05935-f008:**
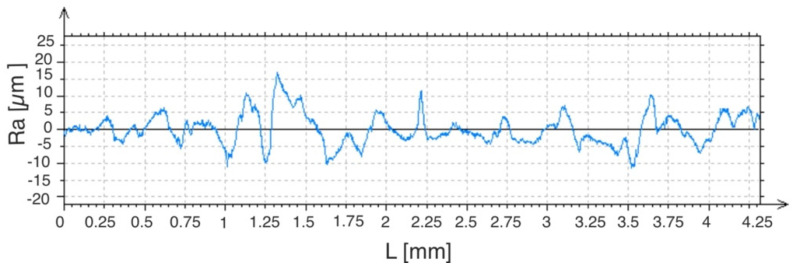
Examples of measurement results of the WC-Co-Al_2_O_3_ coating microgeometry parameters after laser treatment.

**Figure 9 materials-15-05935-f009:**
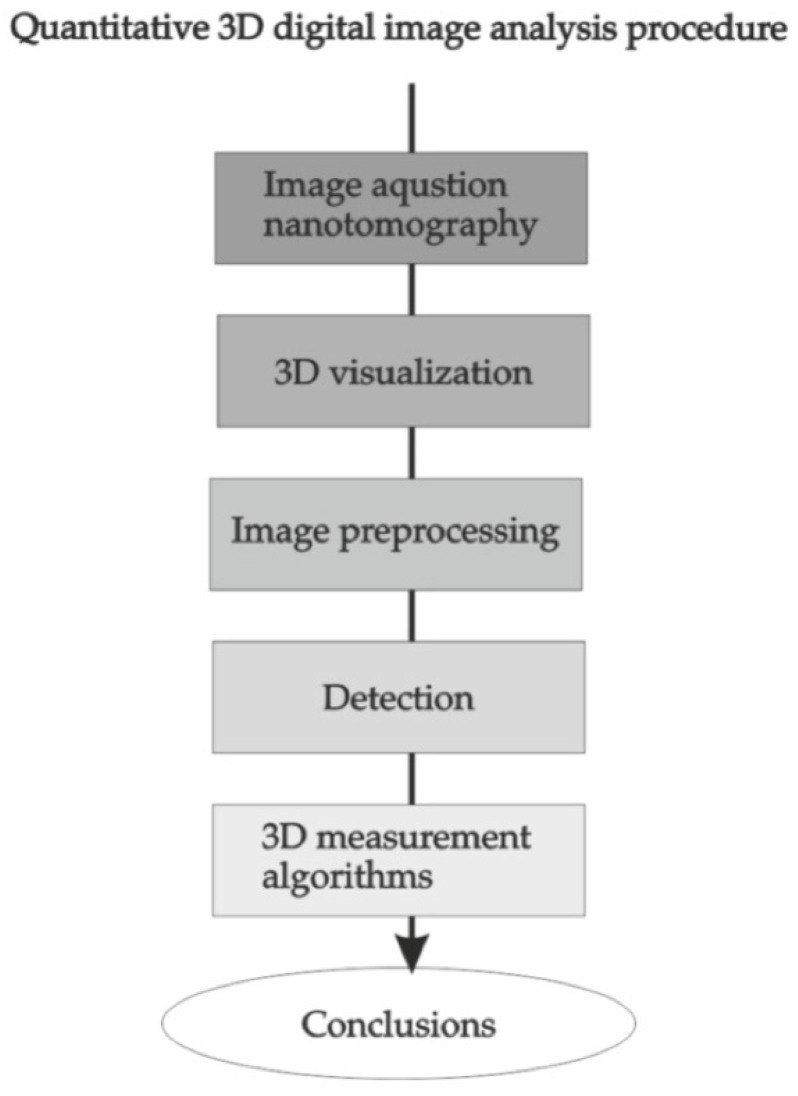
Scheme of the procedure for quantitative analysis of three-dimensional images.

**Figure 10 materials-15-05935-f010:**
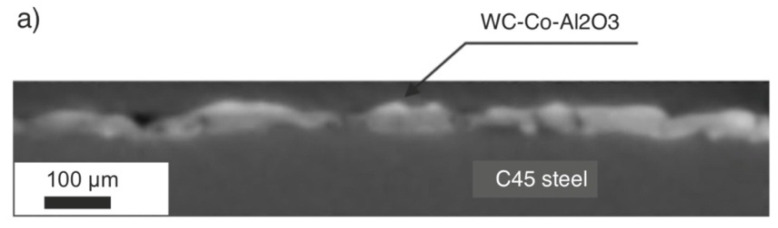
Perpendicular cross-section view of tested samples. (**a**) Example slice of 3D image sample before laser treatment. (**b**) Example slice of three-dimensional image sample after laser treatment.

**Figure 11 materials-15-05935-f011:**
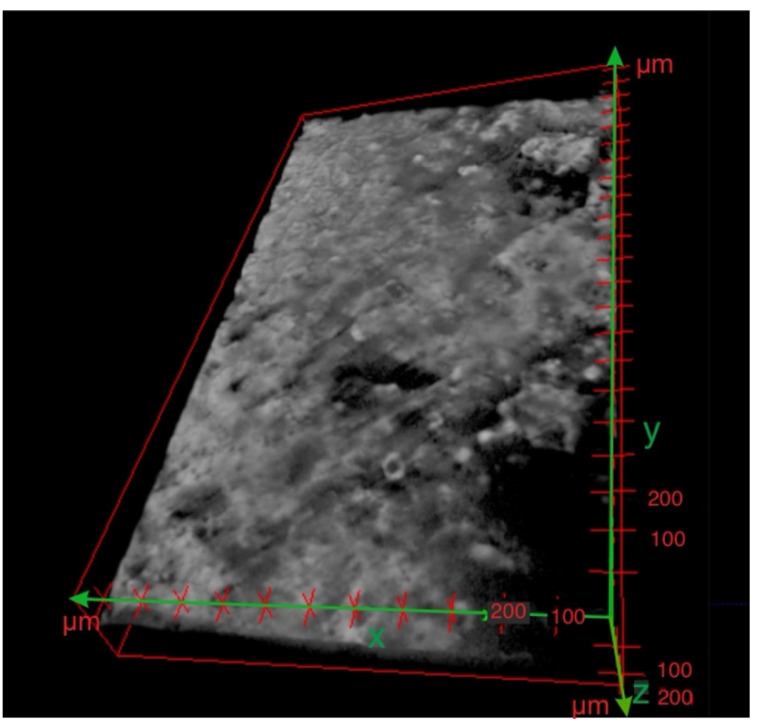
Three-dimensional reconstruction of the nano CT scan of the specimen before laser treatment.

**Figure 12 materials-15-05935-f012:**
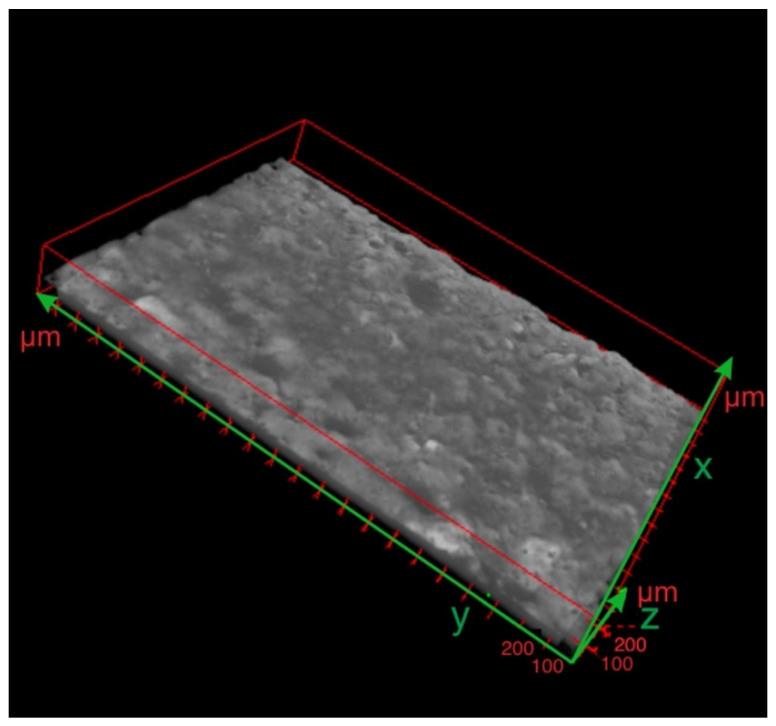
Three-dimensional reconstruction of the X-ray nano CT scan after laser treatment.

**Figure 13 materials-15-05935-f013:**
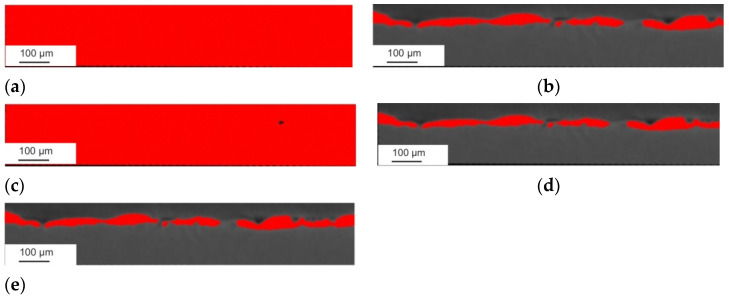
Detection of the coating layer on an example layer of the specimen before laser treatment” (**a**) Li method, (**b**) maxEntropy, (**c**) Otsu method, (**d**) Yen method, and (**e**) RenyiEntropy method.

**Figure 14 materials-15-05935-f014:**
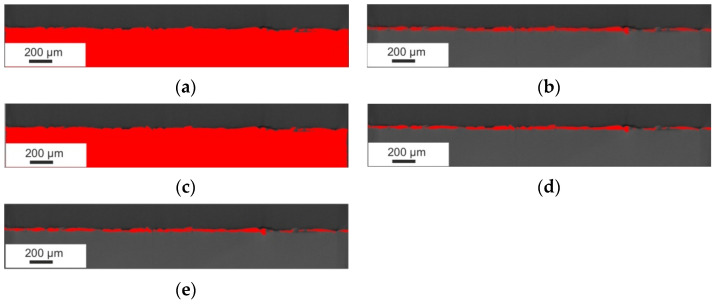
Comparison of the detection accuracy of different automatic threshold algorithms on an example layer of the specimen after laser treatment: (**a**) Li method, (**b**) max entropy method, (**c**) Otsu method, (**d**) Yen method, and (**e**) RenyiEntropy method.

**Figure 15 materials-15-05935-f015:**
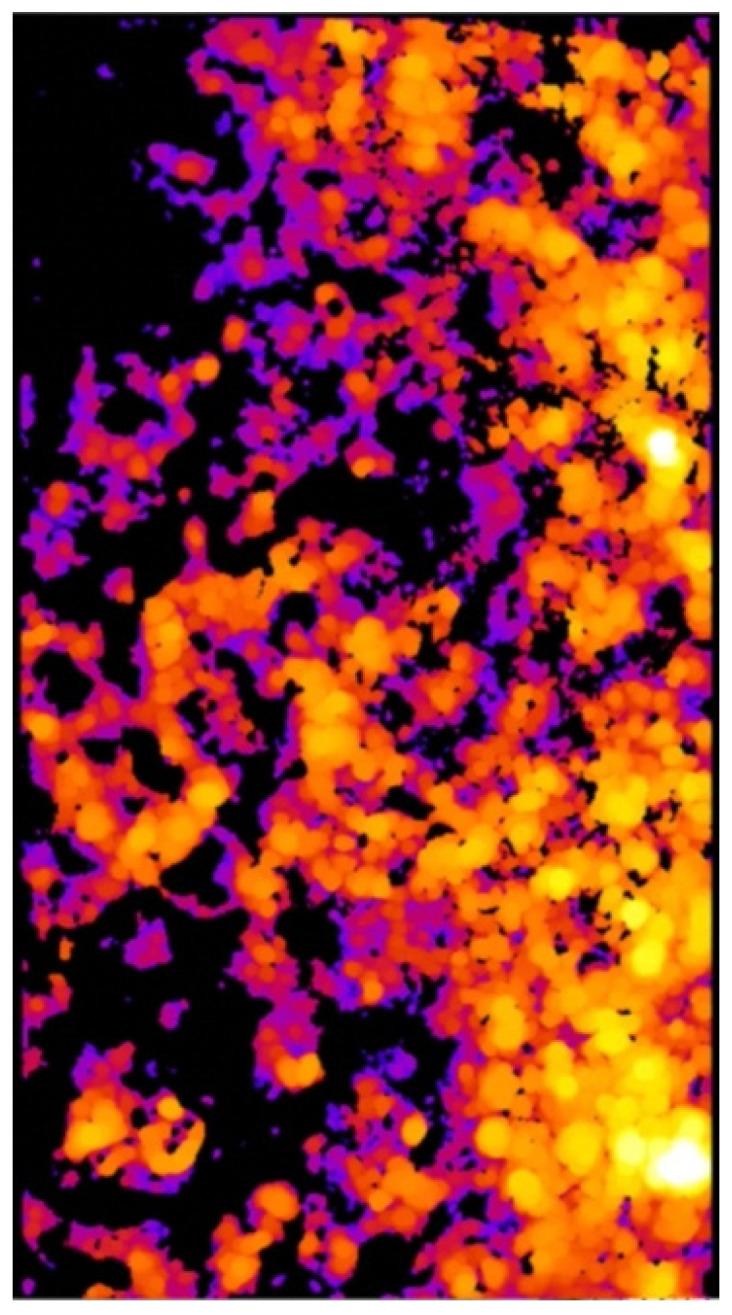
Thickness map of the coating before laser treatment.

**Figure 16 materials-15-05935-f016:**
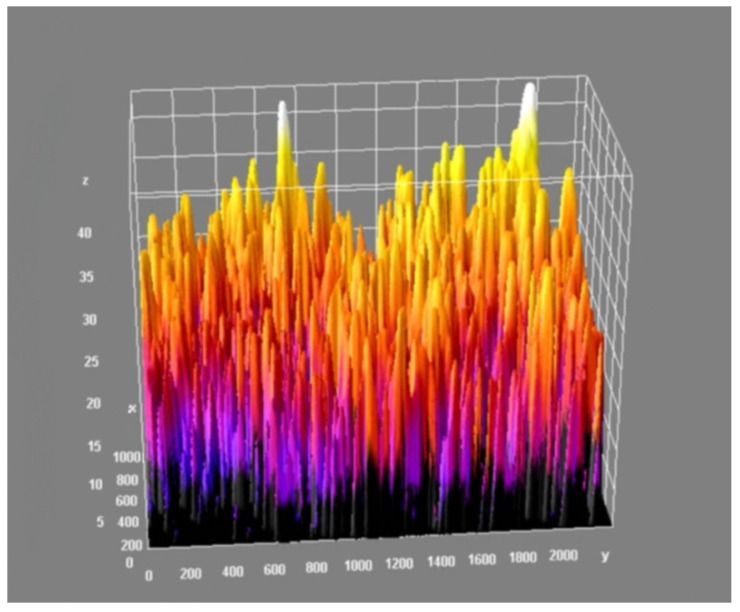
Three-dimensional chart of the thickness layer on the analysed volume of the coating before laser treatment.

**Figure 17 materials-15-05935-f017:**
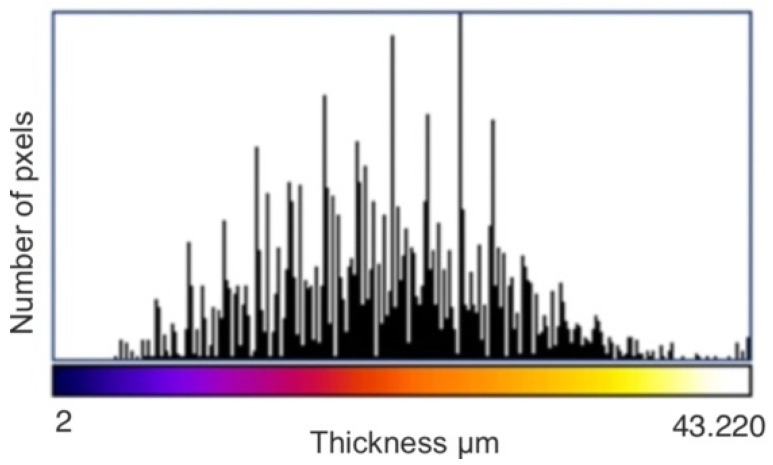
Histogram of the thickness of the ESD coating before laser treatment.

**Figure 18 materials-15-05935-f018:**
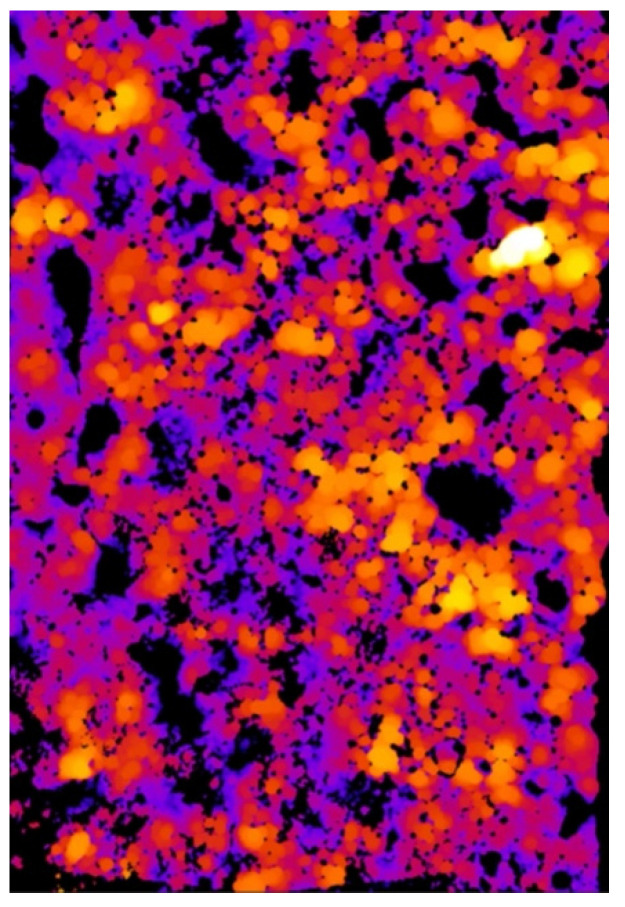
Thickness map of the coating after laser treatment.

**Figure 19 materials-15-05935-f019:**
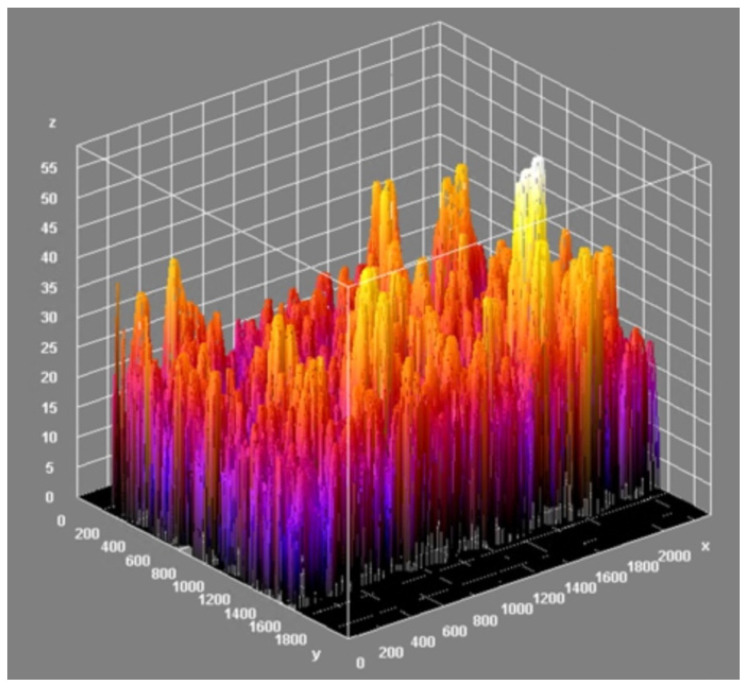
Three-dimensional chart of the thickness layer on the analysed volume of coating after laser treatment.

**Figure 20 materials-15-05935-f020:**
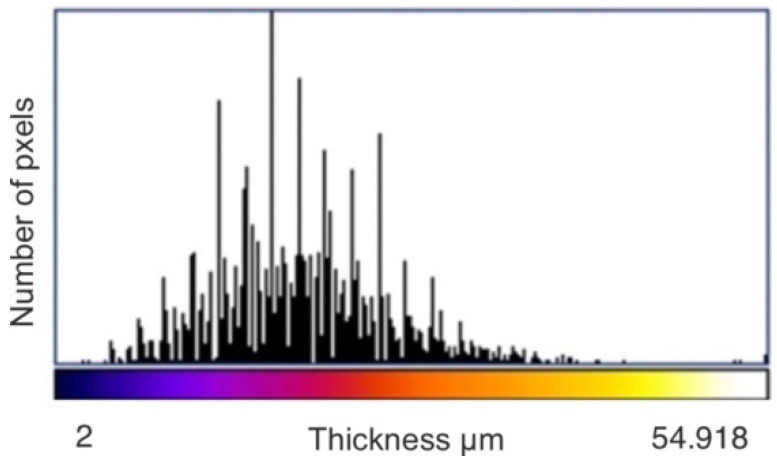
Histogram of the thickness of the ESD coating after laser treatment.

**Table 1 materials-15-05935-t001:** Parameters of the SGS of the topography of WC-Co-Al_2_O_3_ coatings before and after laser beam machining.

Parameters	Coating	
	WC-Co-Al_2_O_3_	WC-Co-Al_2_O_3_ + LBM
*Sp* (μm)	32.24	31.93
*Sv* (μm)	21.80	21.64
*Sz* (μm)	54.01	53.77
*Sa* (μm)	3.94	3.66
*Sq* (μm)	5.17	5.07
*Ssk*	0.18	0.87
*Sku*	4.24	7.20

**Table 2 materials-15-05935-t002:** Results of quantitative analysis of the cermet coating.

Sample	N Number of Objects	V Total Volume (μm^3^)	S Total Surface Area (μm^2^)	N_V_ (1/mm^3^)	S/V (1/μm)	A_A_ (%)
Before laser treatment	136	30,976,965	4,627,679	4390	0.14	70%
After laser treatment	80	53,077,663	10,121,956	1507	0.19	95%

**Table 3 materials-15-05935-t003:** Results of the thickness measurements.

Sample	Mean (μm)	Min (μm)	Max (μm)	Standard Deviation (μm)	Mode (μm)
Before laser treatment	21.731	2	43.22	6.88	25.99
After laser treatment	20.488	2	54.92	6.94	18.02

## Data Availability

The data presented in this study are available on request from the corresponding author.

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
