# Peer review of "Nano X-ray Tomography Application for Quantitative Surface Layer Geometry Analysis after Laser Beam Modification"

_materials, 2022, doi:10.3390/ma15175935_

Round 1
Reviewer 1 Report
The paper presents the nano X-ray tomography application for the quantitative surface layer geometry analysis after laser beam modification. The final results shows that 3D analysis significantly enriches information about the geometry of the surface layer, providing data which are unable with other research techniques. The authors need to address the following issues/comments for publications by the journal:
1. The abstract lacks stating the gap that is not filled using the available literature and this research is going to fill. Please briefly describe the unsolved problem that you've solved with the presented research. In addition, it should be shortened, as an abstract is not a summary that contains all the work in the paper – but rather one that analyses and showcases only the critical research contributions and the best results.
2. Also, the organization is poor relatively and I recommend that you reorganize the paper for a good quality structure. For example, if there is only one separate section (Section 2.1), it is recommended not to separate it.
3. The organization of Section 2 is poor too. Only microstructure analysis is given, while the nano X-ray tomography testing parameters, which is the most important testing method in this paper was not given first in this section of Materials and Methods. Maybe section 3.4.1 should be in the section of Materials and Methods.
4. Figure 1 and Figure 2: Figure 1 (a) and Figure 1 (b) maybe more appreciated.
5. The lack of scale bars, especially for CT imaging results, is academically extremely lax. Please recheck Figure 1, Figure 10-14.
6. Similarly, for data graphs, there should be axes headings (like Figure 7-8, Figure 17 and Figure 20). The font should be clear.
7. Giving key conclusions in the form of lists would be more appropriate for the Conclusions section.

Author Response
Thank you for your precious time and efforts invested in reviewing this manuscript. We appreciate the time and effort you and the reviewers have dedicated to providing valuable feedback on my manuscript, and we are grateful for your insightful comments on my paper. We have highlighted the changes within the manuscript.
Please find in the attached file our responses to your comments.

Reviewer 2 Report
The manuscript by Gądek-Moszczak and coworkers studies how post laser treatment affects the structural properties of a cermet coating layer over steel. By means of XRD, SEM, optical profilometry and 3D XRay imaging they show that the surface coverage and quality is improved by the laser treatment; in particular, they identify a set of parameters that can be extracted from X-Ray nanotomography and allow a quantitative comparison of the material before and after the treatment.
The work is interesting and correctly presented, the improvement provided by the laser treatment emerges clearly from the available data. There are only a few minor concerns in presentation which are listed in the following:
* some lines in materials and methods are repeated twice (158-168)
* Standard techniques such as XRD and profilometry should be presented also in materials and methods
* Some parameters are not clearly derfined, particularly Sa. Clarify SGS quantities
* Explanation of thresholding is not very clear (372-385)
*Spelling of Renyi keeps changing in the text
* The values presented in tables 1 and 3 are quite close for before and after the laser treatment. If possible error estimation should be added, otherwise a comment to what constitutes a significant difference should be added in the text.
Author Response
Thanks for your precious time and efforts invested in reviewing this manuscript. We appreciate the time and effort that you and the reviewers have dedicated to providing your valuable feedback on my manuscript. We are grateful for your insightful comments on my paper, and we have highlighted the changes within the manuscript. In the attached file, we prested our responses to your comments.

Reviewer 3 Report
See attached review.

Author Response
Thanks for your precious time and efforts invested in reviewing this manuscript. We appreciate the time and effort that you and the reviewers have dedicated to providing your valuable feedback on my manuscript. We are grateful for your insightful comments on my paper, and we have highlighted the changes within the manuscript.
Please find in the attached file our responses to your comments.

Round 2
Reviewer 1 Report
Accept in present form